# ROYAL SOCIETY
# OPEN SCIENCE

Reciprocity and alignment: quantifying coupling
in dynamic interactions. *R. Soc. Open Sci.* **8**:
210138.

behaviour/cognition/neuroscience

coupling, reciprocity, social cognition,
alignment, multi-scale dynamics

**Author for correspondence:**
Merle T. Fairhurst
e-mail: m.fairhurst@lmu.de

# Reciprocity and alignment: quantifying coupling in dynamic interactions

Guillaume Dumas[1,2] and Merle T. Fairhurst[3,4]

[1]CHU Sainte-Justine Research Center, Department of Psychiatry, University of Montreal,
Quebec, Canada
[2]Mila – Quebec Artificial Intelligence Institute, University of Montreal, Quebec, Canada
[3]Institute of Psychology, Faculty of Human Sciences, Bundeswehr University, Munich, Germany
[4]Faculty of Philosophy and Munich Center for Neuroscience, Ludwig Maximilian University,
Munich, Germany

GD, 0000-0002-2253-1844; MTF, 0000-0001-6540-5891

Recent accounts of social cognition focus on *how* we do things
together, suggesting that becoming aligned relies on a reciprocal
exchange of information. The next step is to develop richer
computational methods that quantify the degree of coupling
and describe the nature of the information exchange. We put
forward a definition of coupling, comparing it to related
terminology and detail, available computational methods and
the level of organization to which they pertain, presenting
them as a hierarchy from weakest to richest forms of coupling.
The rationale is that a temporally coherent link between
two dynamical systems at the lowest level of organization
sustains mutual adaptation and alignment at the highest
level. Postulating that when we do things together, we do so
dynamically over time and we argue that to determine and
measure instances of true reciprocity in social exchanges is key.
Along with this computationally rich definition of coupling,
we present challenges for the field to be tackled by a
diverse community working towards a dynamic account of
social cognition.

## 1. Not what but how we do things with others

Beyond simply doing something together, what makes our
interactions with other social agents interesting, appealing and
useful must surely depend on *how* we do things together. As
such, a recent theoretical account of alignment shifts the focus
from the nature of the task to the nature of the exchange of
socially relevant information [1]. This dynamic interplay
between self and other, a certain give-and-take, results in the
mutual and reciprocal adaptation of our behaviours to

communicate, understand and coordinate with one another. Recent research in psychology and neuroscience has investigated the simultaneous and coordinated activity of two minds (brains) when they were 'aligned', 'coupled' or 'synchronized'. Physiological, neurological and behavioural patterns of coupling have been reported, coupling which varies depending on whether the individuals are involved in truly reciprocal ongoing interactive dynamics with others or merely infer, or simulate, the content of others' minds at a distance [2,3]. In the following, we try to clarify the terminology used in the literature (table 1), describe types of coupling in terms of a theoretical hierarchy (table 2), and compile the available computational methods allowing for clear distinctions to be made between instances of inter-agent interaction (table 3).

Consider the three cases depicted in figure 1. A keen amateur dancer watches his favourite celebrity dance pair on one of the many dancing shows on television—he imagines himself in their shoes as they glide across the floor (figure 1a). This describes an instance in which social cognition may be required but in which there is clearly no information exchanged between the amateur watching his TV and either of the dancers on screen. He may, taken with the music, entrain a foot tap or a shoulder shrug in time with the beat, an example of so-called physical alignment. Here again though, there is only a unidirectional flow of information. Let us shift to the celebrity pair dancing, let us say, a salsa (figure 1b). The two dancers are coordinating their intricate, showy moves to the rhythmical music, each individually vying for the limelight. In this case, one could describe this as a coupling between each dancer and the external, musical timekeeper. Of course, this may be in addition to some degree of inter-dancer coupling. We and others [18] might posit that if, by contrast, the pair were dancing the Argentinian tango, an infinitely more intimate style of dance which requires of the pair to mutually adapt to each other's moves, one would assume the degree of coupling between the two dancers to be richer and greater.

In §2, we first describe the term reciprocity and then review the existing social cognition literature in which dynamic tasks are used and in which the concepts of reciprocity and coupling are discussed. The tasks might take richer forms of social interaction such as dance, as described above or reduced models of cooperation like the extensive literature on sensorimotor synchronization (e.g. synchronized finger tapping). The measures of behavioural alignment described will range from studies of mutual eye gaze to temporal synchronization as well as additional measures of neural coupling intended to bolster a quantification of the degree of coupling present. This review will be organized by describing different levels of coupling, spanning from spurious synchronization to linguistic alignment, as a function of the degree of information exchanged (table 2). This will delineate current confusion in the field as to what coupling entails and distinguish between types of coupling.

In §3, we then propose key challenges for cognitive science to further study coordinated interaction in humans. This will include a discussion of the current limitations of analysis techniques for the investigation of coupling, with an overview of current measures used in the literature and potential ones in development (table 3). This also means, based on the synthesis of tasks and guided by how the available technology and analysis methods and models allow us to observe and quantify coupling, we propose potential avenues for future work. Specifically, with a theoretically and computationally richer description of coupling, we present four areas that can be extended, namely going beyond coupling (e.g. uncoupling and metastability), going beyond the dyad (i.e. larger groups), designing computational models and social machines and developing experimental tasks that cut across the levels of coupling.

## 2. Reciprocity and coupling

We have suggested that rather than focus on *what* individuals are doing together, we should instead attempt to describe and quantify *how* they are interacting; that is, detail the nature of the exchange of information between interacting agents [1]. Moreover, we have suggested that as a spectrum and as a degree of the reciprocal information flow, interactions can be graded as more or less social. Interestingly, this quantitative approach as a function of information flow is also used for describing reciprocity in non-biological systems as well [19]. In humans, this type of referential communication and evidence for mechanisms that support this ability to infer the intentions of others and compare self and other can be seen from as early as two months of age [20]. It has, however, been suggested that these capacities must be fine-tuned throughout development [21,22]. This conceptualization of reciprocity has been looked at in dyads [23] and groups [24] both between human–human interactions but also those involving human and non-human agents [25,26].

**Table 1.** Glossary of terminology.

| term | definition |
| --- | --- |
| adaptation | adjustment to behaviour in response to perceived social cues in order to coordinate. |
| alignment | the dynamic and reciprocal adjustment of the components of a system for its coordinated functioning; at the social level, it can refer to the state of agreement or cooperation among persons or groups. Reciprocal adjustment can be asymmetrical. |
| brain-to-brain | can refer to two different concepts: (i) the technological communication from one brain to another by directly extracting signal from one and stimulating the other according to certain rules or (ii) the actual coupling of neural processes in one brain to the neural processes in another brain via the transmission of a signal through the environment. |
| chaotic itinerancy | 'universal dynamics in high-dimensional dynamical systems, showing itinerant motion among varieties of low-dimensional ordered states through high-dimensional chaos'. |
| cooperation | the process of multiple organisms acting together for common or mutual benefit, as opposed to working in competition for selfish benefit. |
| coordination | the process of organizing components of a system so that they work together properly and well. It is characterized by stable relative timing of the movement components. |
| coordination dynamics | theoretical approach to explain and predict how patterns of coordination form, adapt, persist and change in living things. In coordination dynamics, components of a system communicate via mutual information exchange (cf. coupling) and information is both meaningful and specific to the forms coordination takes. |
| coupling | two systems are said to be coupled when they are interacting with each other. The coupling often refers to the relational strength. |
| emotional contagion | phenomenon of having one person's emotions and related behaviours directly trigger similar emotions and behaviours in other people. |
| empathy | the ability to understand and share the feelings of another. |
| entrainment | the synchronization of a single or multiple systems to an external rhythm. |
| extended cognition | view of cognition that considers mental processes going beyond the body to also include aspects of the environment and the organism's interaction with that environment. |
| handshaking/negotiation | a term used in computing to describe the exchanging standardized signals between devices in a computer network to regulate the transfer of data. |
| imitation | advanced behaviour whereby an individual observes and replicates another's behaviour. |
| information flow | when measurable quantities of one system depend on those of another. |
| joint action | ability to coordinate our actions with those of others to achieve a shared goal. |
| mimicry | the tendency to copy gestures and facial expressions of others. Mimicry is thus to repeat something, albeit not necessarily accurately. In this sense, it can also be seen as a superficial means of imitation. |
| mutual influence | used in the developmental psychology literature to describe patterns of interactive regulation between infant and caregiver. |
| prediction | in tightly coupled systems that interact together dynamically over time, one might assume a high degree of prediction of a partner's behaviour allowing for greater and smoother coordination. |

| Term | Definition |
| --- | --- |
| reciprocity | the quality of an exchange with mutual dependence, action, or influence. At the sensorimotor level, it may refer to the back-and-forth flow of perception and action during social interaction and, at a more representational level (e.g. social psychology and economics), may refer to the symmetrical aspect of rules and reciprocal treatment a person can give back in function of what they have received. |
| second-person neuroscience | conceptual and empirical approach to the investigation of social cognition focused on second-person engagements, related to the feelings of engagement at the emotional level, and the intricate reciprocal relations with others through social interaction. |
| signalling | used in computing, economics and neuroscience, where in each case it generally describes the exchange of information between involved points/agents in the network. |
| social machine | hybrid systems governed by both computational and social processes. |
| strategic communication | communicating information/signalling (in a dynamic task, this may take the form of behavioural adaptations) that is helpful for coordination by allowing more efficient target prediction. |
| symmetry | describes the nature of the exchange or the underlying information being exchanged which may or may not be balanced across interacting agents. In an asymmetric exchange, not all participating individuals have access to the same amount or type of information. |
| synchronization | emergent property that occurs in a broad range of dynamical systems as their temporal alignment. In humans, it is often used to describe coordinated movements in unison, different from mimicry, which refer to similarity at morphological level but can occur with delay. |
| two-body neuroscience | theoretical approach to human socio-cognitive abilities emphasizing both the embodied nature of individual cognition and the reciprocal aspects of social interaction. |
| two-person neuroscience or 2PN | term introduced by Riitta Hari to push forward the study of brain functions in two persons at the same time (in contrast with 1PN). It is thus different from second person neuroscience referring to different perspectives (i.e. first person and third person). |

**Table 2.** Hierarchy of coupling. Summary of theoretical hierarchy of levels of coupling. For each level of coupling described in §2, we summarize what distinguishes one level from the previous as well as describing what information is exchanged, how this information exchange is studied and how this level relates to cognition. These distinctions are useful in theoretical terms to establish the kinds of information that are exchanged at each level, that is the richness of the exchange, and to identify the best ways to quantify the degree of coupling, that is the appropriate task and computational approaches to use empirically. It should be stressed that these levels do not exist in isolation but as one might expect, one level of coupling may facilitate and indeed lead to a higher level of coupling. In the developmental case, the primary dyad of caregiver–infant may demonstrate physiological coupling in the form of synchronized heartbeats (physiological coupling) which may in turn lead to higher-order means of communication (goal/semantic alignment).

| | differentiable by | what is exchanged | how it is studied | related to cognition |
|---|---|---|---|---|
| spurious coupling | — driven by similarity of input<br>— no information is exchanged<br>— not under conscious control | nothing | — needs to be differentiated from physiological coupling<br>— inter-individual similarity on a given social task (e.g. neurocinematic)<br>— heterogeneity across health and diseases<br>— computational models to estimate the contribution of structure to dynamical similarity | — similarity is nevertheless a pre-requirement for communication at a certain point |
| physiological coupling | — coordinated, though almost certainly unconscious, exchange | — unconscious physiological changes<br>— signal changes in moods/states<br>— often reciprocal | — correlation between physiological measures<br>— joint action paradigms | — can scale up to conscious awareness of coupling<br>— affiliation |
| entrainment | — behavioural/observable<br>synchronized output varies as a function of the coupling of co-actors | — spontaneous<br>— intention or willingness to interact<br>— content of representation may be minimal, temporal components underlying synchronization | — temporal in-phase synchronization<br>— still primarily correlative in nature | — may facilitate more conscious levels of coordination |

(*Continued.*)

**Table 2.** (*Continued.*)

| | differentiable by | what is exchanged | how it is studied | related to cognition |
|---|---|---|---|---|
| sensorimotor coupling | — adaptive and predictive mechanisms that allow for coordination (though not necessarily conscious) | — actions encoded by virtue of temporal and spatial properties of movements<br>— encoded signals may include roles (leader/follower), mental states of the individuals<br>— may also encode properties of target joint attention (indirect object) | — temporal coordination patterns (e.g. sensorimotor synchronization)<br>— patterns indicative of prediction or adaptation strategies<br>— neural correlates<br>— primarily correlative in nature<br>— optimality of synchronization: in some cases synchronization can be orthogonal to the richness of the exchange, that is the degree of coupling | — 'Social-glue'<br>— as a form of non-verbal communication, may be a precursor to language. |
| goal/semantic alignment | — cultural tool to transmit our intentions and goals (prior to full-blown language-based goal)<br>— e.g. gesture<br>— meaning/sense making | — goals<br>— intentions<br>— motivational states/emotions<br>— symbols reference to something that is not present | — developmental transition to language<br>— decision making | — major distinction with other animals<br>— prerequisite for language development |

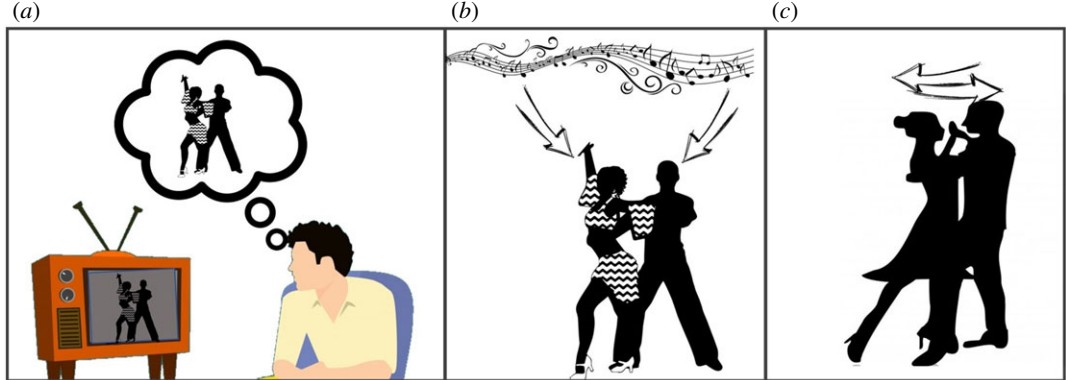

**Figure 1.** Coupling and alignment as a function of an exchange of socially relevant information. Offline observation of dancers on a screen (*a*) may engage social cognition brain networks but this case does not involve a reciprocal exchange of information and as such, other than possible entrainment through coordinated foot tapping in time with the beat, results in little or no coupling between the TV watcher and the dancers on the screen. This would therefore be described as a weak form of alignment. By contrast, the two dancers engaged in a salsa (*b*) individually entrain with the rhythm of the music while interacting with and adapting to each other. This results in a certain level of coupling which can be quantified. Although seemingly similar, the two dancers locked into a tight hold for an Argentinian tango (*c*) may, as a function of a greater degree of information exchange, exhibit higher levels of coupling in this more intimate style of dance that requires tighter coordination between interacting partners.

**Table 3.** Quantifying interactions: a summary of mathematical methods to measure coupling. Many tools have been proposed to quantify coupling but there is no gold standard, as each presents benefits and limitations. Four main features are of matter of interest (as columns): directedness, the ability to attribute directionality to the coupling on top of its strength; linearity, the fact that the coupling is proportional to the change of the inputs; complexity, the required burden in computations necessary to obtain the measure (i.e. proxy of computation duration); and stationarity, how the method require core characteristics of the time series (e.g. mean, variance and spectral characteristics) to remain stable relative to the timescale of the analyses. Abbreviations: PDC partial direct coherence; ARX autoregressive with eXogenous; PLV phase locking value; MPD mean phase difference; wPLI weighted phase locking index; CCOR circular correlation coefficient.

| methods | directed | linear | complexity | stationarity | references |
|---|---|---|---|---|---|
| correlation | no | yes | low | yes | [4,5] |
| coherence | no | yes | low | yes | [6] |
| Granger causality/PDC/ARX | yes | yes | high | yes | [7–10] |
| PLV/MPD/wPLI/CCOR | no | no | low | no | [11–13] |
| cross-recurrence | yes | no | low | no | [14,15] |
| transfer entropy/mutual information | yes | no | high | no | [16,17] |

Perhaps most generally, reciprocity is the interdependency between units of a coupled system [27] and describes accommodation and adaptation that allow two or more agents to become aligned. Importantly, this process begins with perception of the nature of an information exchange. Studies have shown that from a third person perspective, we are able to extract details of the exchange so as to imagine and understand interactions that do not involve ourselves [28,29]. According to Sebanz *et al.* [21], this process in which agents infer the intentions and adapting to the action of others depends on shared representations of objects and tasks, shared attention and the ability to integrate the predicted effects of one's own and others' actions. The ability to create shared representations more generally is potentially dependent on mirror neuron system (MNS) which facilitates action anticipation. In addition, de Bruin *et al.* suggest the need for perspective taking [22]. Most relevant to the sections that follow, however, are the quantifiable accommodation patterns of reciprocity, that is ways to measure the manner, level and degree of the interdependency of interacting agents. The diversity of these patterns as a function of context mean that the flow can but does not necessarily have to be symmetrical [30] and may involve either mirrored or complementary responses [21,31].

Based on growing literature describing the ways in which individuals coordinate in time and space, we have compiled words used to refer to and describe reciprocity in human interaction (table 2). A challenge remains in clearly assessing the differences and commonalities between these terms beyond their origin and the phenomenon they are supposed to describe. At the community level, the term 'coupling' may be the least connoted and thus we choose to use it as a common thread.

In the following section, we clarify through examples how the word coupling takes different flavours across the literature and may account for different phenomena. Specifically, by the order in which the examples are presented, we describe a hierarchy, organized from weakest to strongest (or richest) levels of coupling (table 1 and table 2). In the examples presented, we describe not only the nature of the information exchanged (if any) but also describe the kind of coupling this might produce along with the ways this coupling has or may be quantified.

## 2.1. Similarity, spurious coupling and shared input

Different factors can contribute to the observation of a temporally coherent link between two dynamical systems. Before even trying to decipher the way in which these systems might interact together through different levels of coupling, spurious ones must be discussed. Here, the temporal correlations observed do not correspond to a coupling between the two systems—or at least at the timescale of interest—but to a lack of independence between them. Their dependence can come from shared external perturbation or a common intrinsic property. The major risk would be to draw incorrect inferences, so beware of 'the spectre of 'spurious' correlations' [30].

Shared noise may be the most common source of dependence between two signals. What is usually considered as background noise may include uncontrolled perturbations spanning across physical, physiological and even psychological levels. At the physical level, this includes the environmental electromagnetic noise (e.g. power line at 50 or 60 Hz) or even thermal noise inducing common physiological responses (i.e. sweating). The physiological level is particularly sensitive for neuroimaging where artefacts like eye blinks, muscles (e.g. smiles) or heartbeat can also create an illusion of neural coupling. Finally, at the psychological level, uncontrolled environmental factors such as sounds heard by participants or shared visual perturbations (e.g. the light if participants are in the same room) may also increase the noise in the data.

Common property is also a classic confounding factor, although this remains more often implicit and thus ignored. Statisticians have warned against the inflation of correlation by shared non-stationarity, variance or autocorrelation. High non-stationarity could become especially problematic for long-term correlations (note: this phenomenon is especially documented for the unit root stochastic processes) [31]. Those issues are particularly important to keep in mind when investigating social interaction, especially in studies of interbrain 'coupling' [32]. Burgess [11] recently showed how similar spectral modulation by the same task can lead to a spurious increase of synchronization between the brain activity of two participants, even in the absence of any exchange of information.

Fortunately, there are good practices to limit spurious coupling and even techniques to avoid them. Burgess [11] for instance, recommends a focus on 'improved experimental control and the use of a different measure of phase synchronization'. Some measures such as circular correlation (CCor) or weighted phase-locking index (wPLI) are for instance less biased estimators of synchronization than spectral similarity change. It must be noted that wPLi is very specific to magnetoencephalography (MEG) and electroencephalography (EEG) data and is used as an estimate of synchronization between neural signals. Dean & Dunsmuir [32] advise to detrend and 'prewhiten the series being cross-correlated'. Besides correlation, they suggest the use of predictive models (e.g. autoregressive or Granger causality), still admitting that 'causal intervention experiments are commonly necessary to determine whether the model genuinely captures influences at work in the system'.

Spurious coupling can also be studied on its own as an interesting measure of shared contamination by the environment. For instance, in the case of the report by Hasson et al. [33], Burgess [11] states that 'the participants simultaneously experience the same stimuli such as watching a movie together, even though they are not directly interacting'. This can be seen as a false-positive at other levels but studying such similarity can lead to insights about how different people react to similar natural stimuli. For instance, social contexts tend to maximize the correlation of blood oxygen level-dependent (BOLD) signals across individuals looking at the same movie. Such experimental design can also better quantify between-subject variance and how different neural pathways can sustain the same task [34]. In psychiatry, the inter-individual variability is even characteristic of certain disorders, autism being the canonical example. In autism spectrum disorders (ASD), there is a strong heterogeneity at

both biological and phenotypical levels. Some even argue ASD is associated with a higher internal noise and poor external noise filtering [35]. Such structural and dynamical heterogeneity will affect measures of coupling, even spurious ones, but this dissimilarity of coupling could have functional consequences on the propensity to create genuine coupling [36].

## 2.2. Unconscious, physiological coupling

Though spurious physical and neuro/physiological coupling driven by the similarity of input must be qualified and controlled for, it certainly must not be confused with the unconscious but the coordinated coupling of individuals at the sub-personal level. Specifically, as a marked difference from spurious coupling, in these cases, we see that although not under conscious control, changes evoked at a physiological level are coordinated between interacting agents. There is already a wealth of research exploring how behavioural coordination, social cohesion and indeed feelings of affiliation depend on or result in unconscious physiological coupling.

Starting with the primary dyad, infants and their caregivers will typically exchange information in a dynamic manner that will result in a degree of reciprocity and synchrony which begins at a physiological level [37]. Both the synchronization of heartbeats [38] and levels of oxytocin, the so-called 'bonding hormone' [39], have been shown to enhance physiological and behavioural readiness for social engagement. Engagement of these systems has been observed to continue into adulthood, where, for example, in cases of physically coordinated musical groups, coupling of breathing and cardiac rates has been quantified [40]. Additionally, oxytocin is thought to underlie the enhancement of inter-brain synchrony in male adults [41]. Similarly, a priming study on romantic couples identified a correlation between increased accuracy in rating negative emotional states and higher synchrony in their skin conductance and time of pulse transmission from the heart to the fingers [42]. Konvalinka *et al.* [43] have quantified information exchange as a function of 'interpersonal similarity' showing that firewalkers and related spectators (i.e. family members or friends) were more likely than unrelated individuals to become coupled at a physiological level. This unconscious signalling is surely a form of communication which, through physiological changes, signals changes in mood or state (though not explicitly). Whether through coordination of heartbeats or respiration rate, information is exchanged in order to initiate or facilitate alignment. The factors that modulate physiological coupling are still largely unknown; however, recent work on interpersonal touch has shown that interpersonal respirational and heart rate coupling is increased during partner touch [44]. Moreover, this new line of evidence shows that the affective context (i.e. the presence of pain) modulates the effect touch has on physiological coupling.

This kind of coupling is measured, most generally speaking, as a correlation between physiological measures. These methods are constantly being refined with measure-specific approaches allowing one to quantify degrees of synchrony and thus potentially a measure of emotional coherence across interacting agents. Of course, what might be of most interest is that this unconscious coordination at a physiological level may, and in certain cases does, scale up to a level of conscious awareness of coupling.

## 2.3. Spontaneous, unconscious motor coupling

Based on the definition by [2], we suggest that examples such as coordinated rocking or swaying at a concert, or walking in step down a sidewalk represent a primarily physical level of alignment, akin to the kind of coordinated action seen in flocks of birds [45]. Although it is certain that this kind of coordinated and often tightly coupled, temporally synchronized motor behaviour allows a diverse range of species to become a social unit [46,47], these types of alignment are assumed not to be as rich as the consciously coordinated, dynamically adaptive changes we make, say, in group music making. Within this literature, however, coordinated actions are still described as varying in the degree of stability and magnitude [48]. In contrast with what we would assume are richer forms of social interactions, ones which are intentional and where the higher degree of coupling is intrinsic to the task (e.g. rowing), in more spontaneous forms of alignment various perceptual-motor couplings result in synchronization at a physical level [49,50]. Specifically, we assume that at this level, some degree of information is exchanged either in the form of or which results in observable synchronous motor output which varies as a function of the coupling of co-actors. As such, the methods used to quantify this generally passive phenomenon are often limited to correlation [51]. An interesting case to consider is *entrainment*, which refers to individuals becoming physically entrained to a common external rhythmic stimulus. In this example, a temporal signal in the music produces a physical,

sensorimotor coupling between the listener and the musical beat. Is anything communicated here? This brings to bear the idea that in these cases of unconscious coupling, one must consider how necessary intention or willingness to interact is to higher levels of coupling.

The development illustrates well how unconscious motor coupling and innate access to others' emotional states [52] can lead to more advanced sensorimotor coupling and higher semantic alignment, especially with language [53]. This transition phase demonstrates how the levels of coupling we are discussing here do not exist in isolation. During development, the physiological coupling may prompt spontaneous entrainment and via feedback loop may then allow interacting agents, mother and child in this case, to move into a more adaptive level of sensorimotor interactions. Those adaptive levels range from the primary sense of agency to the ability to communicate with the other, not only by reproduction of existing forms but also through the creation of new patterns, and in the end the ability to anticipate the behaviour of the other and even mentalize their intention. The sense of agency self-organizes progressively through the coupling with the world [54,55] and the right balance between full autonomy and reactivity [56]. The creativity develops with the emergence of new patterns through co-regulation between the child and its caretakers [57]. This co-regulation starts with imitation of adults, but young children also initiate and maintain similar communicative behaviour with their preverbal peers [58]. Even children with autism, although described as typically deficient in social interaction and often inattentive to others' social behaviours, are sensitive to being imitated [59,60]. Through those imitative, and not necessarily goal-oriented interactions, children build their self–other equivalences for actions which lead them to better anticipate what the others will do [61] and to interpret others as having similar psychological states [62,63]. The shared representation of self and other leading to action experience has been postulated as important for representational understanding and mentalizing [64–66].

Spontaneous and unconscious motor coupling could thus constitute the beginning of the path toward the Theory of Mind [67]. It thus seems there are both qualitative and quantitative differences between these forms of passive motor coupling and both entrainment or imitation. Specifically, as explored in more detail in the next section, examples like coordinated movement on rocking chairs rely on and are triggered by a fairly basic perturbation from the outside. By contrast, higher levels of coupling such as sensorimotor coupling may be initiated by an external stimulus and maintained internally through a higher degree of reciprocal information exchange. In these more active forms of interaction, the reproduced movement may involve a degree of anticipation [68], potentially relying on internal models and memory processes [69] resulting in an altered version of the behaviour and leading to the emergence of new patterns. Additionally, if one was to describe the signal produced, in these highly repetitive motor coupling events, one would observe both a higher degree of rhythmicity, which may be absent say in imitation, longer trains of events (instances of mimicry are typically limited to 3–5 s), and potentially some lag between the two interacting signals.

## 2.4. Sensorimotor coupling

As discussed in the previous section, more spontaneous, unconscious examples of motor coupling (such as temporal entrainment) may communicate the intention or willingness to interact. From the developmental literature, we see that what may start as a spontaneous, internally generated action may result in a cycle of coordinated responses and permit mother and child to move into a more adaptive level of sensorimotor interactions [70]. It therefore seems key to point at this time that this may be an example of a transition phase between levels of coupling; that is, that although presented separately in this present discussion, these levels don't exist in isolation. Through feedback loops, this mechanism becomes a useful strategy to understand and learn about self and the environment.

As one moves conceptually to the level of sensorimotor coupling, we start considering cases in which an external stimulus triggers an appropriate and coordinated response. This is a natural 'joint' extension of within-agent action–perception coupling [21]. Specifically, through links and neural overlap between action planning and perception, both within the individual and joint-action cases, sensorimotor systems allow for both an adaptive and predictive coordination between perceived sensory stimuli and an appropriate motor response [71]. I hear an interesting beat, I anticipate the onset of the next beat and I tap my foot in rhythm with it. In a joint-action scenario, I see you clap your hands, I predict the onset of the next beat and I clap my hands in synchrony with you. It should be noted that beyond external triggers in the auditory and visual domain, our temporally coordinated actions either in direct physical contact with others (e.g. dancing) or when manipulating an indirect object together with others [72] (e.g. moving a heavy table) may involve coupling that relies on our other senses,

especially that of touch [73]. The deaf percussionist, Dame Evelyn Glennie, coordinates her musical movements at exceptionally precise time scales through sight and tactile vibrations. Empirically, this type of temporal coordination is studied under the umbrella term of sensorimotor synchronization [74]. Whether investigating reduced models of coordination in which participants synchronize finger taps with pacing tones or flashes or richer tasks employing adaptive (and predictive) partners, this vast literature demonstrates a higher level of coupling between the two signals. An important point of clarification must be made at this point, namely that one should not confuse observed synchronization either at the level of behaviour or at the level of the brain in which two correlated signals simply follow the same pattern in time with true coordination in which two signals are coupled as a function of adaptive and predictive mechanisms.

Sensorimotor coordination differs from the previous level of coupling in several ways, and the case of dance and group music making neatly illustrates these differences. As mentioned in §1, there are interesting distinctions to be made and to be studied in seemingly similar types of social interactions, such as dancing the tango or dancing the salsa. By comparing two types of dance we see varying levels of reliance on the external timekeeper as well as the degree of coupling between the dancers [1,18]. In the case of two dancers making synchronized movements in time, observed and even measured synchronization between their movements may or may not be a result of true reciprocity or coupling. One may also speak of the directionality of the exchange of information and the alteration of one's behaviour in response to the perceived stimulus. One player initiates a rallentando, the other may or may not either perceive this cue or slow down sufficiently to coordinate their sounds in time. Sensorimotor coordination can therefore be more or less adaptive and predictive [66]. In lower levels of coordination, we may merely be trying to copy or follow an external stimulus (a fellow agent) as a model but in more complex cases, like group music making, we must both adapt our behaviour to coordinate as well as implement predictive mechanisms to account for more complex tempo changes [75].

From the sensorimotor synchronization literature, one finds a diverse array of methods to quantify temporal coordination and sensorimotor coupling, from estimating the strength of serial dependencies between successive asynchronies during paced finger tapping with a metronome [76] to the coupling between players in a string quartet ([77,78], see §3). This work has provided insight into both the adaptive and predictive mechanisms that underlie coordination during sensorimotor synchronization tasks. From the adaptive side, error correction estimates have been obtained by fitting models to asynchrony time series (for a review, see [78] and used as a proxy for the degree of coupling (described as such by [1,79,80]). Looking more towards the predictive aspect of temporal coordination, using temporal data from the inter-tap-intervals (ITIs from the human tapper) and inter-onset-intervals (IOIs of the pacing signal), Pecenka & Keller [81] used the ratio between the lag-0 and lag-1 cross-correlations of ITIs and IOIs (a prediction-tracking PT ratio) as a measure of prediction in sensorimotor synchronization with tempo changing tapping tasks [81]. Based on several studies, it has been shown that a PT-ratio larger than 1 reflects an individual's tendency to predict tempo changes, while a ratio smaller than 1 indicates a tendency to copy (track) tempo changes. The PT-ratio has been found to correlate positively with musical experience, tapping abilities and neural activation in brain networks comprising cortico-cerebellar motor-related areas and medial cortical areas [82]. Extending the initial (adaptive) correction models, van der Steen & Keller [68] employed simulation techniques to create and test the adaptation and anticipation model (ADAM) of sensorimotor synchronization which incorporates both reactive and predictive elements.

The degree and manner of information exchange may vary as a function of the roles played by the interacting individuals. As investigated in the sensorimotor synchronization literature as well as in richer, real-world examples of coordinated behaviour, 'leaders' (temporal or hierarchical) may set a given tempo or example of behaviour and adapt minimally, while 'followers' will focus their attention on copying and/or following the dictated pattern and adapt more [79,83,84]. How these varying roles are played may be dependent on either mimicking or signalling strategies, with leaders making more communicative actions [85]. The predictability of the 'leader's' behaviour may act as a signal, that serves as a form of non-verbal communication to establish the respective roles of interacting partners [86]. Interestingly, assumedly related to predictability, individuals who are more similar (more 'like-me') perform optimally in sensorimotor synchronization tasks [80].

Non-verbal communication typically starts with mimicry and imitation with many animals imitating and copying the behaviour of their conspecific. This starts early in life, with the co-regulation of exchanges between mother and infant, and the development of social cognition [57]. This mutual influence continues in adulthood with a spontaneous tendency to imitate [87,88]. A question that remains to be clarified, as indeed across all levels of coupling, is how conscious the process might be.

In the case of mimicry, this typically unconscious tendency to copy differs from entrainment in that it is an active phenomenon: it may initially be triggered by an external stimulus but can continue without it [89].

Although the focus of this review is on studies involving dynamic tasks that involve mutual interactions that develop across time, a further difference between this and the previous level of coupling is that while entrainment is recursive, mimicry and other examples of rhythmical imitation can happen as a one-shot event. As such, different computational methods might be useful depending on the number of exchanges that occur within an interaction, with phase-based methods as described or cases of dynamic, rhythmic coordination and information theory measures for single-event behaviour (table 3). In either one-shot or more dynamic cases of imitation, the independence between the stimulus and the imitated response suggests both differentiated neural mechanisms that allow for this ability as well as the need for more sophisticated anticipatory computational methods to quantify coupling in these interactions that go beyond measures of correlation. Specifically, one might assume measures of transfer of entropy as superior to Granger causality estimation since its estimation is more general (nonlinear and non-Gaussian) [90]. From a clinical perspective, a great deal of work continues to be done studying deficits in autism to advance our knowledge of sensorimotor coupling, that is more adaptive reciprocal exchanges. Using coupled oscillator modelling and a pendulum imitation task, this report describes the deficit in social synchronization as a function of coupling [91].

## 2.5. Goal and semantic alignments

Goal-oriented awareness is the ability to perceive goals and perceptions of others; it can range from gaze following and shared attention up to communication of cues and representation [92]. Goal-directed behaviours are complementary and provide a key element of prospective control [93,94]. During development, this ability to infer intentions and attribute goals to others is intrinsically tied to motor cognition [95]; however, there seems to be a chicken-egg problem in what appears first: the ability to interact with others, or the ability to represent them [96]. Grossmann [97] has provided evidence that infants are equipped from birth to preferentially direct their attention to and process social stimuli.

The emergence of meaning starts well before the emergence of language. As mentioned in the previous section, sensorimotor coupling is an interface between the non-verbal and the verbal, the motor and the social, the individual and the collective. The scientific literature illustrates this tension at both theoretical and experimental levels [74]. From human evolution to child development, proper coupling at the sensorimotor level seems the pre-requirement for language. Sensory-motor couplings with the environment stabilize the dynamics of early neural assemblies and thus shape 'neural attractors' [98]. The landscape of spontaneous activity is then able to influence behaviour through those attractors, shifting the organism from passive entrainment to an active coupling [99]. The more those attractors are entrenched, the better they resonate with ongoing coupling. Such phenomena are well documented in other fields (e.g. odour perception) where the resonance of neural dynamics in accordance with past experiences has been proven to encode meaningful events [100].

Many animals coordinate the movement of their bodies, but humans expand this ability to thoughts, including those that we express verbally [101]. Since this alignment of our understanding of the world with the others may be essential to learn and to adapt, there may be strong evolutionary pressure on moving from imitation to language [102]. Vygotsky explains the way in which learners develop their conceptual capacities, working just outside their independent capacity, relying on the supports or scaffolds of their learning environment. For instance, language is considered as initially rising like a means of communication between the child and the people in his environment. This is only later, with the development of internal speech that it comes to organize the thought of the child [48]. There is a lot of similarity with the hypothesis of Michael Grazziano that, evolutionarily speaking, our sense of self has followed the need to interpret the behaviour of others [103]. There is a transfer of the capacity of functional control to language structure and it is possible to demonstrate '[this] continuity of language with other intentional communication by underscoring the richness of the functional organization of co-action that underlies the capacity to use language' [104].

# 3. Beyond traditional coupling

We have seen how social cognition is a braiding of biological, behavioural and social couplings. Based on the synthesis of tasks and methods, we will now delineate some positive proposals for future work: first,

to go beyond the concept of the coupling *per se* by also investigating uncoupling, transient coupling or even metastability; second, to go beyond dyads, through the study of larger groups; third, to better integrate computational approaches, not only for modelling the phenomena but also as social machines integrated into the social interaction itself; and finally, through the development of multi-level experiments, where the intertwined nature of social cognition is probed at all the levels simultaneously.

## 3.1. Beyond coupling: uncoupling and metastability

A good way to understand a phenomenon is to study its opposite. What can uncoupling tell us about coupling? In neuroscience, active desynchronization has been observed [105,106] and may constitute a fundamental mechanism of brain adaptability, with desynchronization preventing the brain from being stuck in a particular state (e.g. epileptic seizure). At the behavioural level, too much synchronization can be a problem (e.g. mob mentality and speculative bubbles), and uncoupling from others can be necessary and adaptive (e.g. end of a musical piece of ensemble music).

Social coordination requires complementary actions, not only pure synchronization. For instance, antiphase coordination at the sensorimotor level already shows a departure from the in-phase mode of coordination. Tackling this aspect, the Haken–Kelso–Bunz model managed to uncover new forms of dynamics and outperformed previous accounts of synchronization focused on the in-phase mode [107]. In dialogues, this is also necessary, and distinctive turn-taking can be observed akin to anti-phase correlated oscillators [108]. By contrast, brief phases of total desynchronization can also be observed [109] thus showing how even the absence of a social sign can become one, signalling boredom or need to take the lead in the interaction. Uncoupling or indeed the shift between phases of being coupled or uncoupled may moreover serve as a signal between interacting agents. Both the fluidity and speed of transition between phases may vary and implicitly communicate a level of expertise [110]. Relatedly, a measure of time to resynchronize, that is the time taken in a temporal coordination task to resynchronize their tapping with the new metre (time to resynchronize, TTR) indicates an ability to disengage from the current entrainment process and to entrain to a new metre [111].

## 3.2. Beyond the dyad: larger groups

Another way of generalizing a principle is to apply recurrence: if $n\_0$ is true, and $n$ implies it works at $n + 1$, then it works, at least theoretically, for any $n$. In this review, we have mostly covered the study of dyadic interactions with only a few studies having ventured beyond the barrier of testing two participants. As Zhang *et al.* [112] put it, there is a blind spot between the 'very few and very many' despite the fact most of our daily interactions take place among larger groups.

Moving beyond the dyad, the types of coupling seen and measured in dyads may act as a mechanism for alignment across larger groups. Richardson *et al.* [48] have shown how individual-level differences in synchrony relate to group-level cohesiveness by analysing movement data with a Kuramoto-based method to quantify cluster phase and investigate patterns of synchrony across six individuals rocking in a circle. Konvalinka *et al.* [43] elegantly quantified dynamic heart rate synchrony between active participants with their related observers, but not with their unrelated observers during a collective fire-walking ritual. Within the musical domain, research has explored these mesoscopic scales looking at small groups of ensemble players [78,113] to the one of a chorus [114]. Again in choruses, oscillatory couplings of cardiac and respiratory activity among singers and the conductor engaged in choir singing have been reported [40]. It is interesting therefore to note that as a function of how these effects are studied, that is through joint action paradigms, coupling at this level outside the laboratory may also stem from common input (i.e. joint attention). In the case of choir singers, studies have explored the manner in which, based on the external timekeeper (conductor) or depending on the audience [114], individuals adjust the intensity of their vocal output in order to optimize the so-called 'self-to-other ratio', which reflects the degree to which an individual can hear their own sounds among co-performers' sounds [115]. Recently, neuroscience even invited itself into the classroom to investigate how a group of students become coupled during learning [116].

Virtual social networks have greatly contributed in the development of mathematical tools to model the larger datasets as related to connectivity between large(r) groups of people. Unfortunately, the focus has been put on static networks rather than dynamical ones. An interesting question to be tackled in future research is whether the degree and richness of coupling naturally have to decrease as a function of interacting agents?

## 3.3. Beyond humans: social machines and coupling with artefacts

The obvious next frontier for the study of social interaction is to investigate the manner in which we coordinate our bodies and minds when we interact with non-human social machines. Technology is increasingly shaping our social structures [117] and we already interact with virtual versions of our loved ones over Skype as well as with artificial agents in the form of video games, automated phone operators, chat bots and hyper frequency trading software (see also, https://www.youtube.com/watch?v=nyJtEGJGkMU). Additionally, scenarios exist and can be imagined in which artefacts couple between each other such as they do for the 'Internet of Things' and between drones. From an academic perspective, the study of inter-agent coupling involving both human and non-human machines allows us to probe further as to the necessary and sufficient criteria and levels of coupling that are required for co-agents to coordinate and become aligned.

Empirically, a great deal of work has already been done, particularly in the domain of temporal coordination, employing social machines or virtual partners to investigate the nature of social interactions. These have ranged from pre-programmed partners providing fixed scenarios for interaction to more adaptive virtual partners [78,79,118,119]. The use of these partners has not only deepened our understanding of coordination behaviour but also to measure changes in emotional responses to either competitive or cooperative conditions when coordinating with the virtual movements of a virtual partner (VP) [120]. In all cases, the use of a social machine is to reliably manipulate the interaction between agents by controlling the VP with programmable algorithms or models which are derived as a function of generalized behavioural dynamics. VP tended to mirror the human's intrinsic behavioural repertoire; a suitable coupling provided the interaction necessary to produce patterns of social coordination. 'The latter were neither the product of the VP's nor the sole outcome of the human's behavioural dispositions, but rather a truly emergent collective pattern that resulted from their interaction' [121].

In general, these social machines can be seen as a dynamical, mathematical mirror where the 'exploration of the machine's behaviour may be viewed as an exploration of us as well' [122]. If artificial machines can serve as a valuable bootstrap of natural machines, the question is how flexible the apparatus must be to deal with co-agents which do not entirely behave, say in terms of richness, as human partners. Moreover, within these mixed agent designs, a particularly interesting question relates to the manner in which goal-directed behaviour is signalled, that is how intentions are communicated between human and machine.

## 3.4. Beyond unitary scale: multi-level experiment and modelling

Since we have demonstrated how multi-scale, our coupling with others can be, the last challenge for future studies is to integrate at least two levels of coupling in one design. Of course, all levels are present by default but here we are considering an explicit experimental protocol allowing the study of two scales and their interaction. A major challenge would be to unwind the cycle of physiological coupling and synchronized behaviour. We may ask 'Which comes first?' but there is not necessarily a defined order associated with these levels of coupling. If no phenomenon occurs after the other but both occur simultaneously, it metaphorically reflects the process of light simultaneously creating shadow [123]. A major challenge remains at both theoretical and methodological levels to nevertheless capture the potential flow of causality between scales [124]. The question of what we measure is thus intimately linked to how we model the whole system and its boundary. Research has mostly focused on how one level predicts or correlates in coupling in another, with those results highlighting local evidence of how levels interact.

If those levels of coupling are happening simultaneously, the coupling may on top account for two major phenomena co-constraining themselves: similarity and communication (figure 2). We have mentioned above that at various levels of coupling, similarity can facilitate reciprocal alignment and communication. However, one might argue that while the degree of coupling may reflect the similarity across the interdependent biological, behavioural and cultural levels, it may also be that similarity is rendered possible thanks to the information exchange (figure 2). Culture illustrates perfectly this double constraint: it is shaped by the never-ending reciprocal exchange between humans, while simultaneously shaping their communication by modulating the similarity of their social environment. But if we take perspective, culture modulates multiple factors in social interaction [125]—the normative approach of cognitive psychology and neuroscience has indeed been questioned by anthropologists [126]. Since culture is shaped through communication between humans, and, as

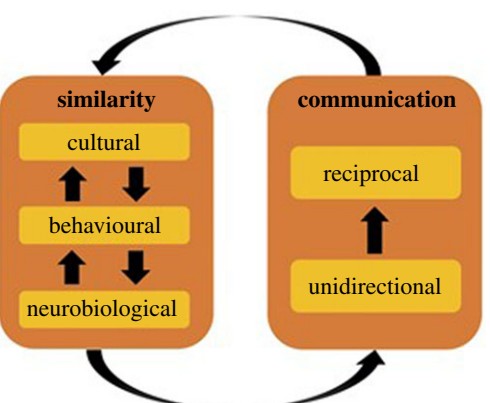

**Figure 2.** Coupling as a measure of similarity and communication between individuals. The observed coupling between individuals measures their active exchange of information through communication, but also their passive similarity across the interdependent biological, behavioural and cultural levels. Interestingly, communication is facilitated between similar individuals, and, simultaneously, similarity is possible thanks to communication, especially at the cultural level.

we argue, the similarity is participating in the facilitation of communication: the two faces of the coin of coupling, i.e. similarity and communication, are like M.C. Escher's Drawing Hands, questioning the classical linear reductionist perspective. Such mutual constraints between being shaped by and shaping are well known in ecology as niche construction. In the case of social cognition, two related challenges are to understand at the phylogenetics level how 'social niches' emerged during evolution [127], especially in *Homo sapiens*, and at the ontogenetic level how individual niche may be disrupted and lead to psychiatric conditions, especially autism [128].

While experiments can provide valuable data for a multi-scale account of social cognition, computational methods have captured the potential mechanisms at play, especially at the scale of neurobehaviour. In an extension to the above section on virtual or social machines, computational social neuroscience has provided human–machine but also machine–machine interaction paradigms. For instance, simulations of two virtual brains interacting have allowed us to probe what may be the role of anatomy in inter-brain synchronization [129]. Numerical models have demonstrated that the shared topology of the human connectome (shaped by evolution) not only contributes to spurious synchronizations but also to the propensity to couple with others through perception-action cycles [130]. While these first results propose new perspectives on how anatomical heterogeneity in autism may contribute to the difficulty of coordinating with others, underlying models need to get more personalized by integrating individuals' anatomy and more realistic biophysical models [131]. Alternative models of interaction have already started to probe social disorders in the context of computational psychiatry [132]. For instance, hierarchical Bayesian modelling has uncovered how social decision is altered in autism [133]. Regression of neural activity based on dyadic behavioural parameters allows to characterize socio-affective phenotypes at the biological level [134,135].

Finally, computational models can apply to group dynamics as well. For instance, Zhang *et al.* [112] have demonstrated how metastable coordination within and between groups is modulated by the diversity of individual preference (e.g. rhythm frequency). Based on Kuramoto and Winfree-based models, a set of specialized prediction-based models aiming to more specifically investigate coordination behaviour in sensorimotor synchronization tasks is under development [136]. These methods will be used to probe both behavioural measures and neural data to quantify the degree of coupling between the interacting agents but also, more importantly, to identify what characteristics (e.g. amplitude, phase) of neurobehavioral time series are involved. Different factors such as the nature of the partner and the nature of the information exchanged can contribute to the observation of a temporally coherent link between the dynamical systems.

## 4. Conclusion

We have seen how our coupling with others is a braiding of biological, behavioural and social coupling, implicating different flavours of what is exchanged between people (and social machines) when they interact. If those levels are of course constructed pragmatically, they also mirror a certain hierarchy of

organizational levels. Overall, there is a tension between the informative nature of varying degrees of predictable signals. Multiple frameworks exist to embrace prediction as the main purpose of integrating information across multiple levels. Predictive coding may be one of the most popular because it provides this integration with a plausible neurophysiological mechanism [136]. Despite the existence of such theories spanning multiple levels, we should remind ourselves about the arbitrariness of those categories. As Claude Bernard said, *'systems are only in the mind of humans'*. This reinforces the need for parsimonious descriptions and concepts that are measurable and applicable at different scales.

Data accessibility. This article has no additional data.
Authors' contributions. Shared first authorship with equal contributions.
Competing interests. We declare we have no competing interests.
Funding. G.D. is funded by the Institute for Data Valorization (IVADO), Montreal and the Fonds de recherche du Québec (FRQ).

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
