## [Peer Review File · Royal Society Open Science]

Review History

RSOS-200221.R0 (Original submission)

Review form: Reviewer 1 (Nicolas Farrugia)

Is the manuscript scientifically sound in its present form?

Yes

Are the interpretations and conclusions justified by the results?

Yes

Is the language acceptable?

Yes

Do you have any ethical concerns with this paper?

No

Have you any concerns about statistical analyses in this paper?

No

Recommendation?

Major revision is needed (please make suggestions in comments)

Comments to the Author(s)

Summary and major comments (major comments will begin with upper case letters)

The paper presents an overview of the field of coupling in dynamic interactions between humans, with a particular focus on the existing methods that enable to quantify such interactions.

The introductory section attempts at a proper definition of "alignment" by considering the social relevance of exchanged information during interactions. Such a consideration of alignment allows to define a hierarchy of interactions, which is nicely demonstrated in Figure 1, rendering the overall point very clear for the reader.

Section 2 consists in a review of the social cognition literature with a focus on task requiring dynamic interactions, and with varying levels of complexity, from "basic" sensorimotor synchronization tasks, up to rich interactions such as dance or joint music making.

(A) Table 2 is introduced in the beginning of Section 2 by the authors, as a "compilation" of words used to refer to and describe reciprocity, but Table 2 actually has a lot more information than that. Table 2 introduces a hierarchy of coupling types, together with a theoretical hierarchy of levels of coupling. However there is no other mention in the main text to Table 2, other than to the "compilation of words" as written at the beginning of section 2.

(B) In addition, some entries in Table 2 seem like they have been left for discussion, or were not finalized? Such as: "Computational models // contribution of structure to dynamical similarity" (the use of the "//" is odd here), or "Intention or willingness to interact??" (same here, the double question mark is odd), as well as "Hasson's work on cinema" (which appears a bit too informal). The formatting of this table also appears quite odd. In addition, such statements, especially the last column (related to cognition), would require a reference to prior studies.

Overall, it seems like Table 2 could be greatly improved, and by doing so it would become a really valuable resource for the reader!

The rest of section 2 is very fluid and well written. Of particular interest is the initial focus on spurious coupling (2.1), it is notable that the authors have made the effort to explain its relevance, the drawbacks as well as the necessity to consider it to some extent (by considering similarity and shared input). The next subsection (2.2) reviews unconscious physiological coupling, by considering physiological signals and neural signals, investigating to which extent such signals can show synchrony across individuals. Subsection 2.3 moves on to review spontaneous forms of motor coupling, and the authors make a very interesting link with ToM at the end of the subsection. The next subsection (2.4) is an extensive review of the sensorimotor coupling literature, and constitutes one of the main novelties of this paper, as to my knowledge the dynamic interactions literature and the SMS / action perception literature have seen relatively few intersections. Finally, subsection 2.5 brings a more advanced perspective on higher levels of interaction, by considering Goal & Semantic alignments, two less-known aspects of this literature.

Section 3 moves on to introduce new concepts for future studies, such as uncoupling and Metastability. However:

(C) I was a little surprised that before moving on to such future perspectives, the authors did not really explain which computational measures were used in which case, with respect to the reviewed literature in section 2. Such an endeavour is supposed to be done by combining the information from Table 2 and Table 3, but as explained in previous comments, Table 2 is not

really described in the main text, and Table 3 is never cited. So I believe there is lack somewhere in the paper to put these different metrics more in context of the reviewed literature.

(D) About Table 2 : references should be added and acronyms should be spelled out. The distinction between Stationary and Dynamical methods would be better explained if there was a dedicated section in the main text as well. It sort of looks as if this table was prepared for the manuscript but not really explained.

(E) Figure 2 is also never mentioned in the paper! And this is quite disturbing because it is quite an ambitious point that the authors are trying to make. This would deserve probably a short paragraph, as well as a few references to back up the proposed "model".

The last two points are particularly important, as I was quite surprised when reaching the end of the manuscript ; the "promise" given in the highlights "We provide a comprehensive comparison of key computational methods to measure reciprocity in human social interaction.", is not fully met because of the abovementioned limitations.

However, the last part of the paper is really innovative and provides very interesting avenues for future work, by considering quite 'provocative' accounts on how to study (un)coupling, on coupling in large groups, as well as the necessity to investigate coupling with / between artificial agents.

Overall, the paper is very promising and I hope the abovementioned comments will help improving it.

minor comments / typos :

page 2 : a recent theoretical account of alignment [1] ... while at page 6, line 27 : Our definition of alignment [1].

Please harmonize, either say "our definition " everywhere or "according to [1]" everywhere.

page 4 : l16 sweeting -> sweating
l18, there is no verb!

page 5 :

l4 : wPLI is very specific to M/EEG, as an estimator of synchronization between neural signals, and an unadvised reader might be confused.

page 6 :

"Such engagement continues into adulthood, where, for example, in cases of physically coordinated musical groups, coupling of breathing and cardiac rates has been quantified. " is this ref [15] ? if not, a reference would be required.

page 7

l18 .[28] This -> the "." should be after [28] and not before

page 8

l18 (such a temporal entrainment) -> such as temporal entrainment

page 9

l2 : "I see you clap your hands, I predict the onset ..." -> this sentence suddenly appears a little too informal when compared to the style of the rest of the paper. Please reformulate.

l6 : The interesting case of the deaf percussionist, Dame Evelyn Glenie, we find someone who coordinates her musical movements at

exceptionally precise time scales through sight and tactile vibrations.

-> the beginning of the sentence needs to be reformulated. "In the interesting case of", and the middle seems a little "off" with regards to the style of the rest of the article (this type of style appears too "oral").

End of page 9, there are two references which appears just after a colon ([43] and [51])

page 16

"If no one comes after the others but both come simultaneously, like light is creating shadow.[100]" -> please reformulate ! there is no verb.

Review form: Reviewer 2

Is the manuscript scientifically sound in its present form?

No

Are the interpretations and conclusions justified by the results?

No

Is the language acceptable?

Yes

Do you have any ethical concerns with this paper?

No

Have you any concerns about statistical analyses in this paper?

No

Recommendation?

Major revision is needed (please make suggestions in comments)

Comments to the Author(s)

This is a very timely systematization. The types of coupling and methods of studying them are indeed multiple, representing different approaches (information processing, dynamical systems). It is indeed a good moment for such a paper.

However this paper can be considered a mere beginning of such work. The concepts and their interrelations are not clearly presented (including 'reciprocity' and 'coupling', 'information flow' and 'information sharing'). The proposed levels also lack clear demarcations and justification.

Sweeping statements are presented on various issues without proper depth and precision (e.g. on culture page 17 lines 11-12; on goal perception, page 12 lines 23-24; on James and Piaget's treatment of infant's abilities to process social stimuli (page 11 lines 28-30, etc), with some threads appearing without clear connections to the discourse (e.g., remarks on odour perception). Last 10 references are missing.

On the other hand, important issues such as movement synchrony vs complementarity in co-action, or dynamical systems novel tools for quantifying coupling such as recurrence analysis are omitted.

I am sorry I cannot be more positive, because I consider the Author's effort definitely well-oriented in the present field of ongoing debates. Yet, a much more in-depth analysis is needed, including the theoretical perspectives from which coupling and reciprocity are studied, in order to achieve a true synthesis and systematization. The paper can definitely be treated as a first step but it would have to be rewritten to be a helpful source for the researchers.

Decision letter (RSOS-200221.R0)

Dear Professor Fairhurst:

Manuscript ID RSOS-200221 entitled "Reciprocity and alignment: quantifying coupling in dynamic interactions" which you submitted to Royal Society Open Science, has been reviewed. The comments from reviewers are included at the bottom of this letter.

In view of the criticisms of the reviewers, the manuscript has been rejected in its current form. However, a new manuscript may be submitted which takes into consideration these comments.

Please note that resubmitting your manuscript does not guarantee eventual acceptance, and that your resubmission will be subject to peer review before a decision is made.

Your resubmitted manuscript should be submitted by 28-Dec-2020. If you are unable to submit by this date please contact the Editorial Office.

on behalf of Dr Giorgia Silani (Associate Editor) and Essi Viding (Subject Editor)
openscience@royalsociety.org

Associate Editor Comments to Author (Dr Giorgia Silani):
Comments to the Author:

We have now received the reviews of your manuscript referenced above. While the reviewers find interest in your data, they have also raised a number of serious concerns, and suggested a major re-writing before publication. These concerns are outlined in their reviews which have been included below. Given the required revisions are consistent, we believe the "reject and allow to resubmit" option to be more appropriate, as it will give you more time to address reviewers' concerns.

Reviewers' Comments to Author:
Reviewer: 1

Comments to the Author(s)
Summary and major comments (major comments will begin with upper case letters)

The paper presents an overview of the field of coupling in dynamic interactions between humans, with a particular focus on the existing methods that enable to quantify such interactions.

The introductory section attempts at a proper definition of "alignment" by considering the social relevance of exchanged information during interactions. Such a consideration of alignment allows to define a hierarchy of interactions, which is nicely demonstrated in Figure 1, rendering the overall point very clear for the reader.

Section 2 consists in a review of the social cognition literature with a focus on task requiring dynamic interactions, and with varying levels of complexity, from "basic" sensorimotor synchronization tasks, up to rich interactions such as dance or joint music making.

(A) Table 2 is introduced in the beginning of Section 2 by the authors, as a "compilation" of words used to refer to and describe reciprocity, but Table 2 actually has a lot more information than that. Table 2 introduces a hierarchy of coupling types, together with a theoretical hierarchy of levels of coupling. However there is no other mention in the main text to Table 2, other than to the "compilation of words" as written at the beginning of section 2.

(B) In addition, some entries in Table 2 seem like they have been left for discussion, or were not finalized? Such as: "Computational models // contribution of structure to dynamical similarity" (the use of the "//" is odd here), or "Intention or willingness to interact??" (same here, the double question mark is odd), as well as "Hasson's work on cinema" (which appears a bit too informal). The formatting of this table also appears quite odd. In addition, such statements, especially the last column (related to cognition), would require a reference to prior studies.

Overall, it seems like Table 2 could be greatly improved, and by doing so it would become a really valuable resource for the reader!

The rest of section 2 is very fluid and well written. Of particular interest is the initial focus on spurious coupling (2.1), it is notable that the authors have made the effort to explain its relevance, the drawbacks as well as the necessity to consider it to some extent (by considering similarity and shared input). The next subsection (2.2) reviews unconscious physiological coupling, by considering physiological signals and neural signals, investigating to which extent such signals can show synchrony across individuals. Subsection 2.3 moves on to review spontaneous forms of motor coupling, and the authors make a very interesting link with ToM at the end of the subsection. The next subsection (2.4) is an extensive review of the sensorimotor

coupling literature, and constitutes one of the main novelty of this paper, as to my knowledge the dynamic interactions literature and the SMS / action perception literature have seen relatively few intersections. Finally, subsection 2.5 brings a more advanced perspective on higher levels of interaction, by considering Goal & Semantic alignments, two less-known aspects of this literature.

Section 3 moves on to introduce new concepts for future studies, such as uncoupling and Metastability. However :

(C) I was a little surprised that before moving on to such future perspectives, the authors did not really explain which computational measures were used in which case, with respect to the reviewed literature in section 2. Such an endeavour is supposed to be done by combining the information from Table 2 and Table 3, but as explained in previous comments, Table 2 is not really described in the main text, and Table 3 is never cited. So I believe there is lack somewhere in the paper to put these different metrics more in context of the reviewed literature.

(D) About Table 2 : references should be added and acronyms should be spelled out. The distinction between Stationary and Dynamical methods would be better explained if there was a dedicated section in the main text as well. It sort of looks as if this table was prepared for the manuscript but not really explained.

(E) Figure 2 is also never mentioned in the paper! And this is quite disturbing because it is quite an ambitious point that the authors are trying to make. This would deserve probably a short paragraph, as well as a few references to back up the proposed "model".

The last two points are particularly important, as I was quite surprised when reaching the end of the manuscript ; the "promise" given in the highlights "We provide a comprehensive comparison of key computational methods to measure reciprocity in human social interaction.", is not fully met because of the abovementioned limitations.

However, the last part of the paper is really innovative and provides very interesting avenues for future work, by considering quite 'provocative' accounts on how to study (un)coupling, on coupling in large groups, as well as the necessity to investigate coupling with / between artificial agents.

Overall, the paper is very promising and I hope the abovementioned comments will help improving it.

minor comments / typos :

page 2 : a recent theoretical account of alignment [1] ... while at page 6, line 27 : Our definition of alignment [1].

Please harmonize, either say "our definition " everywhere or "according to [1]" everywhere.

page 4 : l16 sweeting -> sweating
l18, there is no verb!

page 5 :

l4 : wPLI is very specific to M/EEG, as an estimator of synchronization between neural signals, and an unadvised reader might be confused.

page 6 :

"Such engagement continues into adulthood, where, for example, in cases of physically coordinated musical groups, coupling of breathing and cardiac rates has been quantified. " is this ref [15] ? if not, a reference would be required.

page 7

l18 .[28] This -> the "." should be after [28] and not before

page 8

l18 (such a temporal entrainment) -> such as temporal entrainment

page 9

l2 : "I see you clap your hands, I predict the onset ..." -> this sentence suddenly appears a little too informal when compared to the style of the rest of the paper. Please reformulate.

l6 : The interesting case of the deaf percussionist, Dame Evelyn Glenie, we find someone who coordinates her musical movements at exceptionally precise time scales through sight and tactile vibrations.
-> the beginning of the sentence needs to be reformulated. "In the interesting case of" , and the middle seems a little "off" with regards to the style of the rest of the article (this type of style appears too "oral").

End of page 9, there are two references which appears just after a colon ([43] and [51])

page 16

"If no one comes after the others but both come simultaneously, like light is creating shadow.[100]" -> please reformulate ! there is no verb.

Reviewer: 2

Comments to the Author(s)

This is a very timely systematization. The types of coupling and methods of studying them are indeed multiple, representing different approaches (information processing, dynamical systems). It is indeed a good moment for such a paper.

However this paper can be considered a mere beginning of such work. The concepts and their interrelations are not clearly presented (including 'reciprocity' and 'coupling', 'information flow' and 'information sharing'). The proposed levels also lack clear demarcations and justification.

Sweeping statements are presented on various issues without proper depth and precision (e.g. on culture page 17 lines 11-12; on goal perception, page 12 lines 23-24; on James and Piaget's treatment of infant's abilities to process social stimuli (page 11 lines 28-30, etc), with some threads appearing without clear connections to the discourse (e.g., remarks on odour perception). Last 10 references are missing.

On the other hand, important issues such as movement synchrony vs complementarity in co-action, or dynamical systems novel tools for quantifying coupling such as recurrence analysis are omitted.

I am sorry I cannot be more positive, because I consider the Author's effort definitely well-oriented in the present field of ongoing debates. Yet, a much more in-depth analysis is needed,

including the theoretical perspectives from which coupling and reciprocity are studied, in order to achieve a true synthesis and systematization. The paper can definitely be treated as a first step but it would have to be rewritten to be a helpful source for the researchers.

Author's Response to Decision Letter for (RSOS-200221.R0)

See Appendix A.

RSOS-210138.R0

Review form: Reviewer 1 (Nicolas Farrugia)

Is the manuscript scientifically sound in its present form?

Yes

Are the interpretations and conclusions justified by the results?

Yes

Is the language acceptable?

Yes

Do you have any ethical concerns with this paper?

No

Have you any concerns about statistical analyses in this paper?

No

Recommendation?

Accept as is

Comments to the Author(s)

The authors have addressed all my concerns from the first revision.

Decision letter (RSOS-210138.R0)

Dear Professor Fairhurst,

I am pleased to inform you that your manuscript entitled "Reciprocity and alignment: quantifying coupling in dynamic interactions" is now accepted for publication in Royal Society Open Science.

on behalf of Dr Giorgia Silani (Associate Editor) and Essi Viding (Subject Editor)
openscience@royalsociety.org

Associate Editor Comments to Author (Dr Giorgia Silani):

I am pleased to inform you that your manuscript has been accepted for publication in RSOS. The decision has been based on the positive evaluation of one of the initial reviewers and by my own reading of the paper. We both believe that the manuscript is timely, innovative and provides very interesting avenues for future work. The revised version has now greatly improved and it is ready for publication.

Reviewer comments to Author:

Reviewer: 1

Comments to the Author(s)

The authors have addressed all my concerns from the first revision.

Appendix A

Firstly, may we thank all reviewers for their time and attention to detail in helping us amend and improve our manuscript. We have addressed all comments with responses below (in blue). The major changes to the manuscript are two important paragraphs in section 2, updating the Glossary, re-organisation and improvements in section 3, fully reworked tables 2 and 3 as well as general improvements to style and flow through the text. These can be seen in the track changes document. We sincerely hope that, like us, you now see the merit of this work and thank you in advance for your further consideration.

Reviewer: 1

Summary and major comments (major comments will begin with upper case letters)

The paper presents an overview of the field of coupling in dynamic interactions between humans, with a particular focus on the existing methods that enable to quantify such interactions.

The introductory section attempts at a proper definition of "alignment" by considering the social relevance of exchanged information during interactions. Such a consideration of alignment allows to define a hierarchy of interactions, which is nicely demonstrated in Figure 1, rendering the overall point very clear for the reader.

Section 2 consists in a review of the social cognition literature with a focus on tasks requiring dynamic interactions, and with varying levels of complexity, from "basic" sensorimotor synchronization tasks, up to rich interactions such as dance or joint music making.

(A) Table 2 is introduced in the beginning of Section 2 by the authors, as a "compilation" of words used to refer to and describe reciprocity, but Table 2 actually has a lot more information than that. Table 2 introduces a hierarchy of coupling types, together with a theoretical hierarchy of levels of coupling. However there is no other **mention in the main text to Table 2**, other than to the "compilation of words" as written at the beginning of section 2.

(B) In addition, some entries in Table 2 seem like they have been left for discussion, or were not finalized ? Such as : "Computational models // contribution of structure to dynamical similarity" (the use of the "/" is odd here), or "Intention or willingness to interact??" (same here, the double question mark is odd), as well as "Hasson"s work on cinema" (which appears a bit too informal). The formatting of this table also appears quite odd. In addition, such statements, especially the last column (related to cognition), would require a reference to prior studies.

Overall, it seems like Table 2 could be greatly improved, and by doing so it would become a really valuable resource for the reader !

We thank the reviewer for the positive feedback. We have now corrected all glitches (e.g. "/") in a fully reworked version of Table 2, and make reference to it at relevant points in the text.

The rest of section 2 is very fluid and well written. Of particular interest is the initial focus on spurious coupling (2.1) , it is notable that the authors have made the effort to explain its relevance, the drawbacks as well as the necessity to consider it to some extent (by considering similarity and shared input). The next subsection (2.2) reviews unconscious physiological coupling, by considering physiological signals and neural signals, investigating to which extent such signals can show synchrony across individuals. Subsection 2.3 moves on to review spontaneous forms of motor coupling, and the authors make a very interesting link with ToM at the end of the subsection. The next subsection (2.4) is an extensive review of the sensorimotor coupling literature, and constitutes one of the main novelties of this paper, as to my knowledge the dynamic interactions literature and the SMS / action perception literature have seen relatively few intersections. Finally, subsection 2.5 brings a more advanced perspective on higher levels of interaction, by considering Goal & Semantic alignments, two less-known aspects of this literature.

Section 3 moves on to introduce new concepts for future studies, such as uncoupling and Metastability. However:

(C) I was a little surprised that before moving on to such future perspectives, the authors did not really explain which computational measures were used in which case, with respect to the reviewed literature in section 2. Such an endeavour is supposed to be done by combining the information from Table 2 and Table 3, but as explained in previous comments, Table 2 is not really described in the main text, and Table 3 is never cited. So I believe there is lack somewhere in the paper to put these different metrics more in context of the reviewed literature.

We now have cited Table 2 and 3 in the text and added the most relevant references in Table 3. We believe this will now be a useful tool for researchers in the field. The question of metrics is more associated with the type of data and not necessarily limited to one level. Furthermore, since many people are now trying to capture coupling across several levels, we have aimed to encourage the field towards this multi-scale approach in Section 2.4 (line 288) and 3.4.

(D) About Table 3 : references should be added and acronyms should be spelled out. The distinction between Stationary and Dynamical methods would be better explained if there was a dedicated section in the main text as well. It sort of looks as if this table was prepared for the manuscript but not really explained.

As suggested, we have revised the table 3 by adding references, defining all the acronyms, and explaining what we mean by Stationarity, Dynamic, and Complexity. We believe this format will greatly improve the usefulness of this overview.

(E) Figure 2 is also never mentioned in the paper! And this is quite disturbing because it is quite an ambitious point that the authors are trying to make. This would deserve probably a short paragraph, as well as a few references to back up the proposed "model".

We thank the reviewer for highlighting this shortcoming in our original submission. Additionally, we have made the thread regarding the importance of similarity throughout the manuscript more obvious and include a new paragraph on the relationship between similarity and coupling in a reworking of section 3.4.

The last two points are particularly important, as I was quite surprised when reaching the end of the manuscript ; the "promise" given in the highlights "We provide a comprehensive comparison of key computational methods to measure reciprocity in human social interaction.", is not fully met because of the above mentioned limitations.

See responses to (B), (C), and (D) above.

However, the last part of the paper is really innovative and provides very interesting avenues for future work, by considering quite 'provocative' accounts on how to study (un)coupling, on coupling in large groups, as well as the necessity to investigate coupling with / between artificial agents.

Overall, the paper is very promising and I hope the above mentioned comments will help improving it.

Thank you!

Minor comments / typos :

page 2 : a recent theoretical account of alignment [1] ... while at page 6, line 27 : Our definition of alignment [1].

Please harmonize, either say "our definition " everywhere or "according to [1]" everywhere.

page 4 : l16 sweeting -> sweating
l18, there is no verb!

page 5 :

l4 : wPLI is very specific to M/EEG, as an estimate of synchronization between neural signals, and an unadvised reader might be confused.

page 6 :

"Such engagement continues into adulthood, where, for example, in cases of physically coordinated musical groups, coupling of breathing and cardiac rates has been quantified. " is this ref [15] ? if not, a reference would be required.

page 7

l18 .[28] This -> the "." should be after [28] and not before

page 8

l18 (such a temporal entrainment) -> such as temporal entrainment

page 9

l2 : "I see you clap your hands, I predict the onset ..." -> this sentence suddenly appears a little too informal when compared to the style of the rest of the paper. Please reformulate.

l6 : The interesting case of the deaf percussionist, Dame Evelyn Glennie, we find someone who coordinates her musical movements at exceptionally precise time scales through sight and tactile vibrations.

-> the beginning of the sentence needs to be reformulated. "In the interesting case of" , and the middle seems a little "off" with regards to the style of the rest of the article (this type of style appears too "oral").

End of page 9, there are two references which appears just after a colon ([43] and [51])

page 16

"If no one comes after the others but both come simultaneously, like light is creating shadow.[100]" -> please reformulate ! there is no verb.

All minor typos have been corrected in the updated manuscript. We thank the reviewer for taking the time to highlight these.

Reviewer: 2

Comments to the Author(s)

This is a very timely systematization. The types of coupling and methods of studying them are indeed multiple, representing different approaches (information processing, dynamical systems). It is indeed a good moment for such a paper.

However this paper can be considered a mere beginning of such work. The concepts and their interrelations are not clearly presented (including 'reciprocity' and 'coupling', 'information flow' and 'information sharing'). The proposed levels also lack clear demarcations and justification.

In the updated manuscript, we have added the definition of the terms in the glossary (Table 1) and made revisions to better demarcate the different levels, especially with the new paragraphs at the beginning of Section 2 which include descriptions of information flow, reciprocity and coupling.

Sweeping statements are presented on various issues without proper depth and precision (e.g. on culture page 17 lines 11-12; on goal perception, page 12 lines 23-24; on James and Piaget's treatment of infant's abilities to process social stimuli (page 11 lines 28-30, etc), with some threads appearing without clear connections to the discourse (e.g., remarks on odour perception). Last 10 references are missing.

We removed the mention of James and Piaget. We clarified the reference to attractor resonance in odour perception. Finally, we have updated and checked the references throughout the manuscript.

On the other hand, important issues such as movement synchrony vs complementarity in co-action, or dynamical systems novel tools for quantifying coupling such as recurrence analysis are omitted.

We have now integrated a discussion of the relationship between synchrony and complementarity in co-action with a new paragraph at the beginning of Section 2. We have also added references to Recurrence analysis in Table 3.

I am sorry I cannot be more positive, because I consider the Author's effort definitely well-oriented in the present field of ongoing debates. Yet, a much more in-depth analysis is needed, including the theoretical perspectives from which coupling and reciprocity are studied, in order to achieve a true synthesis and systematization. The paper can definitely be treated as a first step but it would have to be rewritten to be a helpful source for the researchers.

We thank the reviewer for the positive tone of the comment. With the new paragraphs at the beginning of Section 2, we more clearly lay out our theoretical conception of coupling and reciprocity. We believe this revised version better prepares the reader for the major contributions of this paper, namely the systematic detailing of the various levels at which reciprocity and coupling can be studied and quantified.